

**Tracing the sources of dissolved organic carbon occurring in a coastal bay surrounded by**
**heavily industrialized cities using stable carbon isotopes**
Shin-Ah Lee[a], Tae-Hoon Kim[b], and Guebuem Kim[a],*
[a]School of Earth and Environmental Sciences/Research Institute of Oceanography, Seoul
National University, Seoul 08826, Republic of Korea
[b]Department of Earth and Marine Sciences, Jeju National University, Jeju, 63243, Republic of
Korea
*Corresponding author at: School of Earth and Environmental Sciences, Seoul National
University, Seoul 08826, Korea
E-mail address: gkim@snu.ac.kr (G.Kim)



**Abstract**
The sources of dissolved organic matter (DOM) in coastal waters are diverse, and they play
different roles in biogeochemistry and ecosystems. In this study, we measured dissolved
organic carbon (DOC) and nitrogen (DON), $\delta^{13}$C-DOC, and fluorescent dissolved organic
matter (FDOM) in coastal bay waters surrounded by heavily industrialized cities (Masan Bay,
Korea) to determine the different DOM sources in this region. The surface seawater samples
were collected in two sampling campaigns (Aug. 2011 and Aug. 2016). The salinities ranged
from 10 to 21 in 2011 and from 25.4 to 32 in 2016. In 2011, the excess DOC was observed for
higher-salinity waters (16-21), indicating its main source from marine autochthonous
production according to the $\delta^{13}$C-DOC values of −23.7‰ to −20.6‰, higher concentrations of
protein-like FDOM, and lower DOC/DON (C/N) ratios. By contrast, the high DOC waters in
high-salinity waters of 2016 were characterized by low FDOM, more depleted $\delta^{13}$C values of
−28.8‰ to −21.1‰, and high C/N ratios, suggesting that the excess DOC is influenced by
direct land-seawater interactions. Our results show that multiple DOM tracers such as $\delta^{13}$C-
DOC, FDOM, and C/N ratios are powerful for discriminating the complicated sources of DOM
in coastal waters.





**1. Introduction**

Dissolved organic matter (DOM) plays an important role in biogeochemical cycles (e.g., de-oxygenation, acidification, photochemistry) and ecosystems in the ocean (Hansell and Carlson, 2002). DOM composition depends on its parent organic matter and subsequent biogeochemical processes. DOM in coastal waters originates from various sources including (1) *in situ* production by primary production, exudation of aquatic plants, and their degradation (Markager et al., 2011; Carlson and Hansell, 2015); (2) terrestrial sources by the degradation of soil and terrestrial plant matter (Opsahl and Benner, 1997; Bauer and Bianchi, 2011); (3) anthropogenic sources (Griffith and Raymond, 2011). Terrestrial sources are introduced into the ocean via surface runoff, groundwater discharge, and atmospheric deposition.

Depending on the origin and composition of DOM, behavior and cycling of DOM are different: a labile fraction of DOM is decomposed rapidly through microbially or photochemically mediated processes, whereas refractory DOM is resistant to degradation and can persist in the ocean for millennia. In the coastal ocean, organic matter from terrestrial plant litter or soils appears to be more refractory (Cauwet, 2002) and thus behaves conservatively. In addition, refractory DOM is produced in the ocean by bacterial transformation of labile DOM by reshaping its composition (Tremblay and Benner, 2006; Jiao et al., 2010). However, it is still very difficult to determine the sources and characteristics of DOM in coastal waters.

The stable carbon isotopes of dissolved organic carbon ($\delta^{13}$C-DOC) have been used to distinguish different sources. In general, $\delta^{13}$C values derived from C3 and C4 land plants typically range from −23‰ to −34‰ and from −9‰ to −17‰ (Deines, 1980), respectively, while those derived from phytoplankton range from −18 to −22‰ (Kelley et al., 1998; Coffin



and Cifuentes, 1999). In addition, the optically active fraction of DOM known as fluorescent
DOM (FDOM) have been successfully used for characterizing FDOM (Coble et al., 1990;
Coble, 1996). Fluorescence excitation-emission matrices and parallel factor analysis (EEM-
PARAFAC) technique has been applied to trace the source of DOM in many estuaries (Chen
et al., 2004; Jaffé et al., 2004; Murphy et al., 2008; Huang and Chen, 2009). DOC/DON ratios
can also be used to determine the source between allochthonous and autochthonous (Thornton
and McManus, 1994; Andrews et al., 1998; McCallister et al., 2006). The C/N ratios of
terrestrial organic carbon usually are higher than 12, while those of marine organic carbon from
phytoplankton are almost constant ranging from 6 to 8 (Milliman et al., 1984; Lobbes et
al.,2000). Thus, multiple DOM tracers are more powerful for discriminating DOM sources in
coastal waters where various sources are mixed (Faganeli et al., 1988; Wang et al., 2004;
Osburn and Stedmon, 2011; Osburn et al., 2011; Cawley et al., 2012; Pradhan et al., 2014; Ya
et al., 2015; Lee and Kim, 2018).

Thus, in this study, we attempted to use $\delta^{13}$C-DOC, FDOM, and DOC/DON ratios to

differentiate different sources and characteristics of DOM in coastal bay waters surrounded by
heavily industrialized cities.

**2. Materials and methods**
*2.1 Study site*

Masan Bay is located on the southeast coast of Korea with an area of approximately

80 km$^2$ (Fig. 1). The annual precipitation is approximately 1500 mm, most of which occurs in
the summer monsoon season. The amount of freshwater discharge into this bay is
approximately $2.5 \times 10^8$ m$^3$ yr$^{-1}$ with significant seasonal variation. The tide is strong semi-



diurnal, showing a maximum tidal amplitude of ~1.9 m (average = 1.3 m) during the sampling
period. Because of the topographic condition, the current is very weak (2–3 cm s$^{-1}$), and the
residence times of water in the inner bay and in the entire bay are approximately 54 and 23
days, respectively (Lee et al., 2009). This bay is surrounded by cities with thousands of
industrial plants and a population of 1.1 million. This area has been recognized as a highly
eutrophic embayment (Lee and Min, 1990; Yoo, 1991; Hong et al., 2010) in Korea owing to
the massive discharge of sewage and wastewater into the bay. In the middle of the bay, an
artificial island has been constructed since 2015 (Fig. 1) with an area of 0.64 km$^2$. The artificial
island, which is made up of dredged sediments from the bay floor, may result in changes in
water currents, residence times, and biogeochemical conditions.

*2.2 Sampling*

Sampling was conducted in August 2011 and August 2016 in Masan Bay. Water

samples were collected from the surface at 17 sites in 2011 and 10 sites in 2016 from the inner
to the outer bay. The bay receives a large amount of freshwater discharge from the northernmost
part of the region. All water samples were filtered through pre-combusted GF/F filters. Samples
for FDOM analysis were stored at 4°C in pre-combusted amber vials. Samples for DOC, TDN
and δ$^{13}$C-DOC analysis were stored in pre-combusted glass ampoules after acidifying to pH ~2
with 6 M HCL. Samples for dissolved inorganic nutrients (DIN) were stored frozen in a HDPE
bottle (Nalgene) prior to analysis.

*2.3 Analytical methods*

The concentrations of DOC and TDN were determined by using a high-temperature

catalytic oxidation (HTCO) analyzer (TOC-V$_{CPH}$, Shimadzu, Japan). The standardization for



DOC was performed using the calibration curve of acetanilide (C:N ratio = 8) in ultra-pure
water. The acidified samples were sparged with pure air carrier gas for two min to remove
dissolved inorganic carbon. Samples were carried into a combustion tube heated to 720 °C
where the DOC was converted entirely to $CO_2$. $CO_2$ gas was detected by a non-dispersive
infrared detector (NDIR). Our DOC and TDN method were verified by using seawater
reference samples for DOC at 44–46 $\mu$ mol $L^{-1}$ and for TDN at 32–34 $\mu$ mol $L^{-1}$, which were
produced by the University of Miami, Florida, USA. Inorganic nutrients were measured with
nutrient auto-analyzers (Alliance Instruments, FUTURA+ for 2011 samples and QuAAtro39,
SEAL Analytical Ltd. for 2016 samples). Reference seawater materials (KANSO Technos,
Japan) were used for analytical accuracy and verification. DON concentrations were calculated
based on the difference between TDN and DIN concentrations.

$\delta^{13}$C-DOC signatures were determined using a TOC-IR-MS instrument (Isoprime,

Elementar, Germany). The analytical method is the same as Kim et al. (2015) and Lee and Kim
(2016). Low carbon water (< 2 $\mu$M; University of Miami, Hansell's lab) was measured for the
blank corrections and used for preparing all standard samples. The blank correction procedure
used a method previously described (Panetta et al., 2008; De Troyer et al., 2010). Certified
IAEA-CH6 sucrose (International Atomic Energy Agency, −10.45 ± 0.03‰) was used for
standardization. The standard solution was measured for every ten samples to monitor drifting
effects. Also, our measured values of $\delta^{13}$C-DOC for the Deep-Sea Water Reference (University
of Miami) samples fall within ±0.3‰, relative to the values provided by Panetta et al. (2008)
and Lang et al. (2007).





FDOM was determined by using a spectrofluorometer (FluoroMate FS-2, SCINCO)
within two days from collection. Emission (Em) spectra were collected between 250 and 500
nm at 2 nm intervals at excitation (Ex) wavelengths between 250 and 400 nm at 5 nm intervals.
The daily Milli-Q water signals were subtracted from each sample value to remove Raman
scattering peaks. All data were converted to ppb quinine sulfate equivalent (QSE) by using a
quinine sulfate standard solution in 0.1N sulfuric acid at Ex/Em of 350/450 nm. EEMs-
PARAFAC was performed by a MATLAB R2013a program using a DOMFluor toolbox. We
did not correct EEM data for inner filter effects before measurement, since inner filter effects
were found to be negligible for estuarine water samples using this instrument (Lee and Kim,

2018).


**3. Results and Discussion**
*3.1 Horizontal distributions of DOM*
The salinity of surface seawater in Aug. 2011 ranged from 10 to 21, while the salinity in
Aug. 2016 ranged from 25 to 32 (Table 1 and Fig. 2). The concentrations of DOC in both
sampling periods ranged from 100 μM to 200 μM (Fig. 2), which fall within the DOC ranges
commonly observed in coastal waters (Gao et al., 2010; Osburn and Stedmon, 2011; Kim et al.,
2012). The highest concentration of DOC in 2011 was observed at station M4-1 in the middle
of the bay, whereas the highest concentration of DOC in 2016 was observed at station M1,
which is the innermost station in this bay. DOC concentrations were lowest at the outermost
stations in both sampling periods. Concentrations of DON were in the range of 7–24 μM in
2011 and 3–15 μM in 2016, showing the highest value (24 μM) at M5-1 in 2011 and at M1 in
2016 (Fig. 2).



EEMs dataset analyses identified three components for the surface water samples.
Based on the excitation-emission peak location, Component 1 (FDOM$_H$, Ex = 320–360 nm,
Em = 420–460 nm) is associated with a humic-like component (C peak) shown by Coble (2007).
Component 2 (FDOM$_P$, Ex = 275–300 nm, Em = 340–360 nm) is associated with a tryptophan-
like component (T peak), which is a product of microbial processes. Component 3 (FDOM$_M$,
Ex = 290–320 nm, Em = 370–420 nm) is associated with a marine humic-like component (M
peak). We use component 1 as a representative of humic-like FDOM (FDOM$_H$) in this study
since there was a good correlation ($r^2$ =0.95) between component 1 and component 3.

FDOM$_H$ is known to indicate humic substances from terrestrial, anthropogenic, or
agricultural sources (Coble, 2007), whereas FDOM$_P$ is likely related to anthropogenic and
autochthonous sources (Coble, 1996; Hudson et al., 2007). The intensities of FDOM$_H$ and
FDOM$_P$ in 2011 were in the range of 3.6–9.2 ppb QSE and 4–79 ppb QSE, respectively (Fig.
3). The intensities of FDOM$_H$ and FDOM$_P$ in 2016 were in the range of 2.7–0.6 ppb QSE and
4.8–2.1 ppb QSE, respectively (Fig. 3). FDOM$_P$ concentration in 2011 was exceptionally
higher at station M4-1 (78 ppb QSE) relative to that of other stations (2–25 ppb QSE) (Fig. 4d).

*3.2 Origin of excess DOM*
The plot of DOC against salinity in 2011 showed two different mixing trends. The first
slope showed a slight increase in DOC with decreasing salinity toward the innermost stations,
including M1, M1-1, and M2 (Fig.4a, Group 1). The source of DOC in these lower salinity
stations appears to have originated from land by natural and/or anthropogenic processes. The
second trend showed a sharp rise in DOC (excess DOC in 2011) to the maximum between
salinities of 18 and 22 (Fig.4a, Group 2), indicating that there are other DOC sources at the



higher-salinity stations, except for the two end-member mixing. The excess DOC
concentrations in higher-salinity waters could have been produced *in situ* by biological
production and/or from land sources such as sewage treatment plant (STP). The plot of DOC
against salinity showed that DOC in 2016 was in a range similar to that of 2011, although the
influence of fresh water was much less (Fig. 4a, Group 3). This plot shows that there was an
addition of DOC (excess DOC in 2016) for high-salinity water in the bay. The excess DOC
sources in this period could be from either biological production or land-seawater interactions.
In order to determine the main sources of these excess DOC using $\delta^{13}$C-DOC, FDOM, and
DOC/DON ratios, the stations are separated into Group 1, Group 2, and Group 3

Group 1 includes low-salinity water stations (M1, M1-1, M2, M3, M5-1, M5-2, and
M5-3) observed in 2011 (Fig. 1). $\delta^{13}$C-DOC values for Group 1 ranged from −25.4‰ to
−23.3‰. We plotted a conservative mixing curve of $\delta^{13}$C-DOC for the two end-member mixing
equation (Spiker, 1980; Raymond and Bauer, 2001). The end-member values of DOC and $\delta^{13}$C-
DOC were 185 μM and −28‰, respectively, for terrestrial end-member (S=0) and 100 μM and
−18‰, respectively, for marine end-member (S=34). $\delta^{13}$C-DOC values for Group 1 fitted
relatively well into this mixing curve, indicating that DOC for Group 1 was a conservative
mixture of terrestrial C3 land plant (−23‰ to −34‰) in freshwater and open ocean seawater.
$FDOM_H$ in Group 1 was relatively higher than the other groups, with a significant linear
correlation against salinity ($r^2 = 0.89$). This result indicates that humic materials in this region
were mainly from terrestrial sources and behaved conservatively in the course of freshwater
and seawater mixing, which is consistent with previous studies for coastal waters (Coble et al.,
1998; Mayer et al., 1999).



Group 2 includes high-salinity water stations (M4-1, M4-2, M6, M6-1, M7-1, M7-2,
M8, M9, and M9-1) observed in 2011 (Fig. 1). $\delta^{13}$C-DOC values of Group 2 were in the range
of −23.3‰ to −20.6‰, showing much heavier values than the conservative mixing curve.
These values are close to the marine $\delta^{13}$C-DOC values (−22 to −18‰) (Fry et al., 1998) except
for only one station (M6) in this group (−23.3‰). The $\delta^{13}$C-DOC values of Group 2 suggest
that DOM was added *in situ* by biological production in seawater. In contrast to the good
correlation between $FDOM_H$ and salinity for all samples, the concentration of $FDOM_P$ showed
no correlation with salinity. In general, $FDOM_P$ showed non-conservative behavior in many
estuaries owing to the extra source of DOC produced by *in situ* biological production (Benner
and Opsahl, 2001). In the study region, a remarkably high $FDOM_P$ concentration was observed
at station M4-1, where DOC concentration was highest. Thus, this source could originate in
either anthropogenic source inputs and/or *in situ* production (Twardowski and Donaghay, 2001;
Zhang et al., 2009).

Masan Bay has many potential land sources of DOM from different creeks. In addition,
the treated sewage outflow from a STP is located near station M7-1 (Fig. 1). Many studies have
been conducted to identify organic pollutants from STP (Kannan et al., 2010; Lee et al., 2011).
In our study, however, station M7-1 did not show different DOM characteristics: (1) The
concentrations of DOC, $FDOM_H$, and $FDOM_P$ against salinity did not show anomalously
higher or lower trends, relative to the other stations nearby. (2) The $\delta^{13}$C-DOC values at M7-1
(−20.6‰) are close to the marine values, similar to those in other stations nearby. Although
$\delta^{13}$C-DOC values in the sewage treatment plant effluents in this region are unknown, they are
known to be lighter in some US wastewater treatment plants (−26‰) (Griffith et al., 2009). (3)
A fulvic-like peak is not observed, and the increase of $FDOM_P$ intensity at stations M7-1 and



M7-2 was insignificant relative to that at stations M6-1 and M8. $FDOM_P$ is often used as a
tracer of anthropogenic material including treated effluents (Hudson et al., 2007), together with
the fulvic-like peak (Ex/Em 320–340 nm/410–430 nm) (Baker and Inverarity, 2004) which
shows significantly higher values in treated sewage. Thus, we conclude that the concentration
of DOC at station M7-1 was not influenced by STP. This STP appeared to reduce TOC
concentrations to a natural level, as shown in several other estuaries (Abril et al., 2002).
Combining the data of $\delta^{13}$C-DOC and $FDOM_P$ for Group 2, it is likely from marine biological
production, rather than from STP sources.

Group 3 includes high-salinity water stations (M1, M2, M3, M4, M5, M6, and M7)

observed in 2016 (Fig. 1). Although all data were collected in the same wet season (August),
the salinity ranges of both campaigns were different from 2011, with a narrow high salinity
range in 2016. The $\delta^{13}$C-DOC values for Group 3 showed significantly different values relative
to those sampled in 2011 (Group 1 and Group 2). The $\delta^{13}$C-DOC values for Group 3 were
depleted (−28.8‰ and −21.1‰) relative to the conservative mixing curve (Fig. 4b). However,
$FDOM_H$ concentrations were much lower than those in 2011. $FDOM_P$ concentrations were also
lower within a narrow range. These results suggest that DOC in Group 3 is influenced by
terrestrial DOC sources which include lower FDOM. Although the artificial island is made up
of marine dredged sediments, it seems to provide more terrestrial components of DOC and
lower FDOM, perhaps by including land materials that have less humus.

No clear relationships were observed between DON and DOC/DON ratios versus

salinities in 2011. In general, Group 1 showed higher DON (7–24) and lower C/N ratios (6–
21), indicating that the terrestrial DOC from streams included more labile DON, which is





bioavailable. By contrast, Group 3 showed lower DON (4–15) but higher C/N ratios (13–45)
indicating that they include more refractory DON. In general, DOC/DON ratios range from 6
to 8 in the ocean. Thus, these unusually high C/N ratios in DOM seem to be linked to the land
materials that have depleted $\delta^{13}$C-DOC values. The lower DOC/DON ratios in Group 2 seem
to be consistent with DOM production in situ by biological production, as indicated by other
tracers.

**4. Conclusions**

We determine the sources of DOM in 2011 and 2016 using the $\delta^{13}$C-DOC, FDOM, and

DOC/DON ratios. The main sources are separated into three groups based on DOC versus
salinity plots. The DOM in the first group in 2011, which includes the lowest salinity waters,
is found to be a mixture of terrestrial DOM and open-ocean DOM sources based on the $\delta^{13}$C
values of −25.4‰ to −23.3‰ and a good correlation between $FDOM_H$ and salinity. The excess
DOC in the second group in higher salinity waters in 2011 is found to be produced *in situ* by
biological production based on heavier $\delta^{13}$C-DOC values (−22.0‰ to −20.6‰), high $FDOM_P$,
and low C/N ratio. The excess DOC in the third group in high salinity waters in 2016 seems to
be produced by direct interaction between land (the artificial island) and seawater based on
lighter $\delta^{13}$C-DOC values (−28.8‰ and −21.1‰), low FDOM concentrations, and a high C/N
ratio. Our results show that the combination of multiple DOM tracers, including $\delta^{13}$C-DOC,
FDOM, and DOC/DON ratios, is powerful for discriminating the complicated sources of DOM
in coastal waters, which is the critical component of water eutrophication and biogeochemistry.

**Competing interests**
The authors declare that they have no conflict of interest.




**Acknowledgements**

We thank the Environmental and Marine Biogeochemistry Laboratory (EMBL) members for their assistance with sampling and laboratory analyses. This work was supported by the National Research Foundation of Korea (NRF) grant funded by the Korean government (MEST) (NRF-2015R1A2A1A10054309 and NRF-2018R1A2B3001147).

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





**Table 1.** Salinity, DOC, FDOM_H, FDOM_P, and $\delta^{13}$C-DOC in surface water of Masan Bay in
August 2011 and August 2016.

| sampling campaign | station | salinity | DOC | FDOM$_H$ | FDOM$_T$ | $\delta^{13}$C-DOC | DON | DOC/DON |
|---|---|---|---|---|---|---|---|---|
| | | | μM | ppbQSE | ppbQSE | ‰ | μM | |
| Aug. 2011 | M1 | 14.0 | 148 | 6.7 | 13.6 | -25.4 | 12 | 12 |
| | M1-1 | 12.8 | 151 | 9.2 | 14.3 | -24.3 | 7 | 21 |
| | M2 | 10.2 | 157 | 9.0 | 5.4 | -24.6 | 11 | 14 |
| | M3 | 16.3 | 147 | 8.2 | 14.7 | n/a | 16 | 9 |
| | M4-1 | 19.0 | 186 | 7.1 | 78.7 | -21.9 | 13 | 15 |
| | M4-2 | 18.6 | 155 | 6.9 | 8.3 | -21.6 | 10 | 15 |
| | M5-1 | 17.7 | 138 | 4.5 | 4.5 | -23.3 | 24 | 6 |
| | M5-2 | 18.4 | 133 | 5.8 | 20.9 | -24.5 | 11 | 12 |
| | M5-3 | 18.9 | 135 | 8.0 | 11.3 | -23.7 | 13 | 11 |
| | M6 | 18.4 | 146 | 6.6 | 24.8 | -23.3 | 19 | 8 |
| | M6-1 | 19.2 | 142 | 5.5 | 7.4 | n/a | 9 | 15 |
| | M7-1 | 19.5 | 157 | 5.8 | 10.5 | -20.6 | 11 | 15 |
| | M7-2 | 18.9 | 148 | 5.6 | 9.6 | -21.5 | 12 | 12 |
| | M8 | 19.5 | 152 | 5.6 | 7.6 | -21.5 | 15 | 10 |
| | M9 | 18.8 | 149 | 5.6 | 14.5 | -21.9 | 10 | 15 |
| | M9-1 | 19.1 | 154 | 5.1 | 10.2 | -21.0 | 12 | 13 |
| | M9-2 | 20.8 | 106 | 3.6 | 13.1 | -22.0 | 8 | 13 |
| Aug. 2016 | M1 | 29.2 | 191 | 2.7 | 4.8 | -22.8 | 15 | 13 |
| | M2 | 29.9 | 164 | 2.0 | 3.4 | -21.1 | 7 | 22 |
| | M3 | 26.0 | 155 | 2.5 | 3.8 | -28.8 | 8 | 19 |
| | M4 | 27.4 | 149 | 1.9 | 3.5 | -22.6 | 9 | 17 |
| | M5 | 25.5 | 165 | 1.8 | 3.3 | -23.5 | 10 | 16 |
| | M6 | 30.5 | 147 | 1.1 | 3.0 | -23.7 | 6 | 26 |
| | M7 | 31.4 | 166 | 1.3 | 4.4 | -26.2 | 4 | 45 |
| | M8 | 32.0 | 123 | 0.8 | 2.3 | -23.7 | 5 | 26 |
| | M9 | 32.0 | 146 | 0.6 | 2.1 | -24.4 | 5 | 30 |
| | M10 | 31.9 | 130 | 0.7 | 2.7 | -25.0 | 3 | 39 |

n/a = not available.





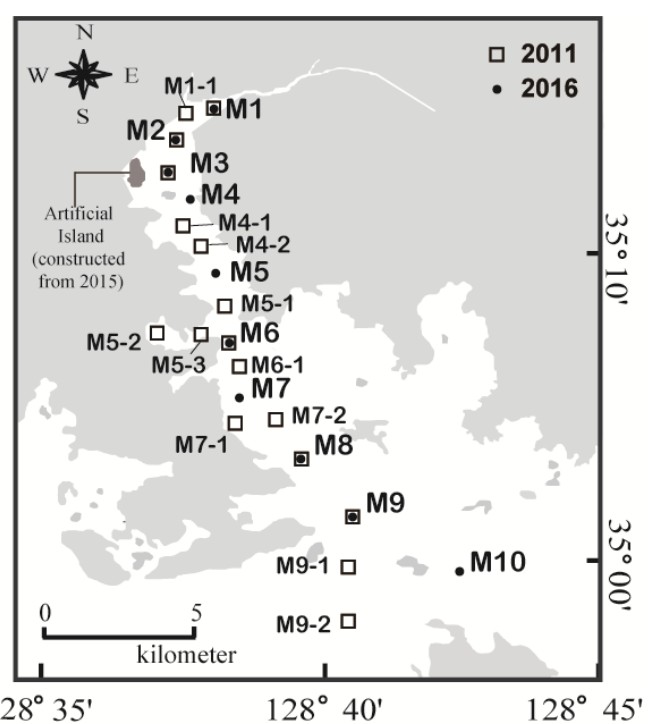


**Figure 1.** A map showing the sampling stations for DOC, δ¹³C-DOC, FDOM, and DOC/DON

ratio in Masan Bay, Korea, in 2011 and 2016.



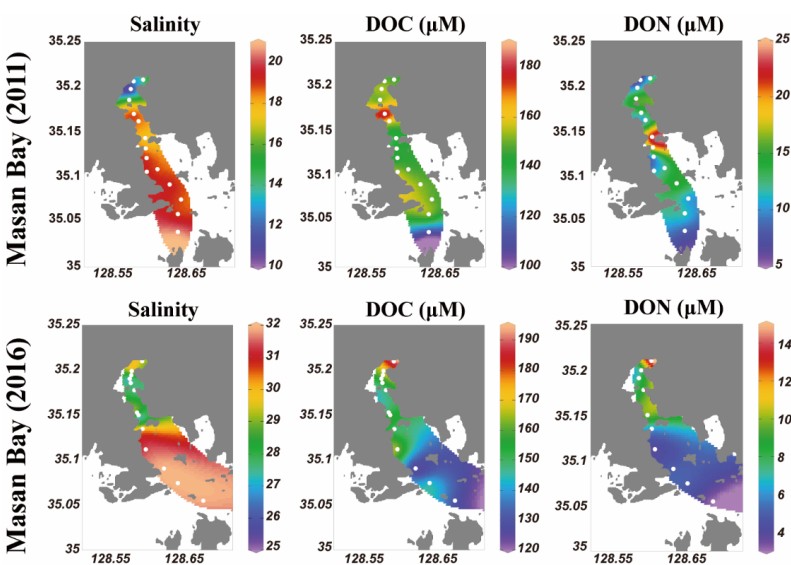

**Figure 2.** Surface distributions of salinity, DOC, and DON in Masan Bay, Korea, in 2011 and

2016.
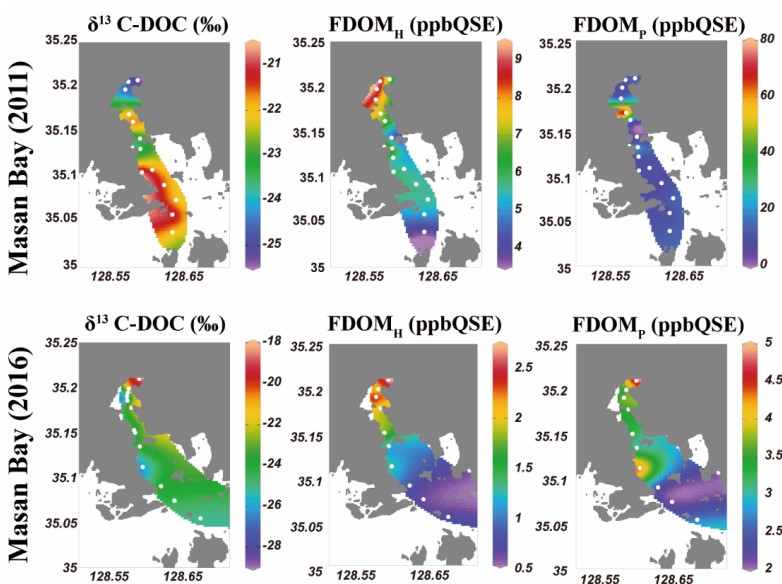

**Figure 3.** Surface distributions of $\delta^{13}$C-DOC, FDOM$_H$, and FDOM$_P$ in Masan Bay, Korea, in 2011 and 2016.

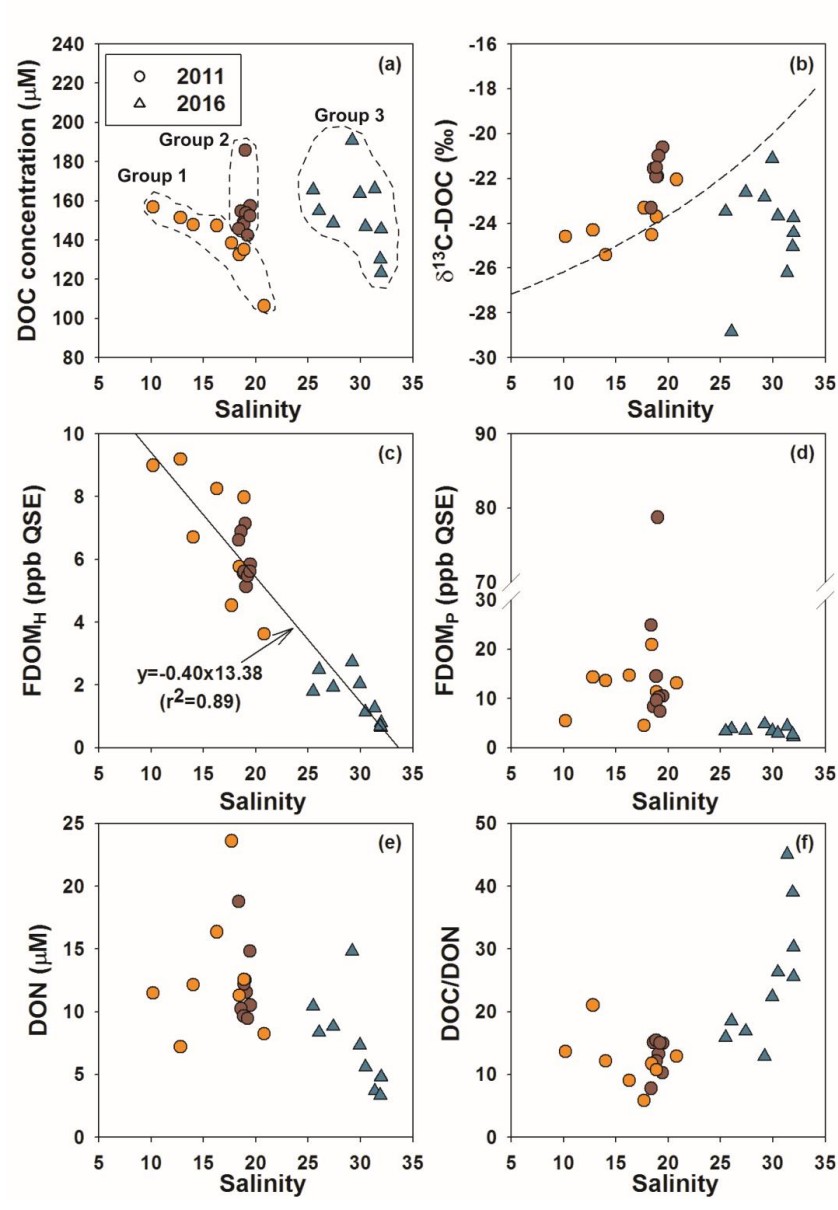

**Figure 4.** Relationships between salinity versus (a) DOC, (b) $\delta^{13}$C-DOC, (c) FDOM$_H$, (d) FDOM$_T$, (e) DON and (f) DOC/DON values. The DOC groups are included in the dashed circles. The dashed line (b) represents the binary conservative mixing line for $\delta^{13}$C-DOC between the terrestrial end-member and the marine end-member.