# Peer review of "Tracing the sources of dissolved organic carbon occurring in a coastal bay surrounded by heavily industrialized cities using stable carbon isotopes"

_Biogeosciences, 2019_

## Referee Comment (RC1) · Anonymous Referee #1 · 6 Aug 2019

This manuscript by Lee et al set out to investigate the sources of dissolved organic matter in a coastal bay surrounded by heavily industrialized cities, Masan Bay, Korea, during two different sampling trips in 2011 and 2016. The authors measured DOC, DON, chromophoric-DOM and stable carbon isotope composition in related samples. They found that the excess DOC was observed in higher-salinity waters (16-21) during 2011, with higher concentrations of protein-like FDOM and lower DOC/DON ratios. In 2016, however, the high DOC waters in high-salinity waters were characterized by low FDOM, more depleted $\delta$13C values and high C/N ratios.

[Figure]

In general, this is a fairly well written manuscript presenting data on the source changes in DOC and CDOM in a coastal environment during two different sampling years. Overall, it is concise manuscript. Their approach is straightforward and conclusions are based on what the authors observed. Contents presented here are suitable for the journal of bg. I support eventual publication of this manuscript

Having said that this manuscript needs revisions before acceptance.

First of all, their two samplings in the same study area are over 5 years apart (2011-2016) without other sampling point. In addition, DOC abundances do not vary consistently or significantly with salinity (Fig 4a) although FDOM-H does (Figure 4c). It is very difficult to judge these DOC data since they are from a coastal bay influenced heavily by industrialized. Additional explanation will be helpful.

Second, the authors presented results from PARAFAC analysis in several places (see also Figure 4) and the FDOM composition differences between two samplings. However, there are not additional information related to these analyses, neither component contours nor excitation/emission loadings. The PARAFAC-derived DOM components should be provided along with their detailed and specific Ex/EM values (can be in supplemental if needed).

Third, Specific goals and scientific questions should be added into the Introduction section. Otherwise, it does not look like a

Fourth, are there any additional parameters/evidence to support the industrial pollution or excess DOC during sampling? Similar to $\delta$13C values, specific C/N ratios should be provided in the abstract so that the reader can easily compare data between different sampling years.

Lastly, similar to DOC concentrations, $\delta$13C values in 2016 are somewhat highly variable within small salinity range. What are the possible reasons causing this abrupt change? FDOM, on the other hand, seemed to vary consistently with salinity. Then,

the question is that are FDOM nor related to the bulk DOC in the study area or from polluted DOC sources?

---

## Author Comment (AC1) · 14 Aug 2019

Lee and Guebuem Kim Anonymous Referee #1

Major comments This manuscript by Lee et al set out to investigate the sources of dissolved organic matter in a coastal bay surrounded by heavily industrialized cities, Masan Bay, Korea, during two different sampling trips in 2011 and 2016. The authors

measured DOC, DON, chromophoric-DOM and stable carbon isotope composition in related samples. They found that the excess DOC was observed in higher-salinity waters (16-21) during 2011, with higher concentrations of protein-like FDOM and lower DOC/DON ratios. In 2016, however, the high DOC waters in high-salinity waters were characterized by low FDOM, more depleted $\delta$13C values and high C/N ratios. In general, this is a fairly well written manuscript presenting data on the source changes in DOC and CDOM in a coastal environment during two different sampling years. Overall, it is concise manuscript. Their approach is straightforward and conclusions are based on what the authors observed. Contents presented here are suitable for the journal of bg. I support eventual publication of this manuscript. Having said that this manuscript needs revisions before acceptance.

=> Thank you for the positive and valuable comments.

First of all, their two samplings in the same study area are over 5 years apart (2011-2016) without other sampling point. In addition, DOC abundances do not vary consistently or significantly with salinity (Fig 4a) although FDOM-H does (Figure 4c). It is very difficult to judge these DOC data since they are from a coastal bay influenced heavily by industrialized. Additional explanation will be helpful.

=> Yes, we will include more explanation on oceanographic changes over different sampling years in the revised version. We use these multiple tracers in order to determine the main sources amongst such inconsistent and dynamic sources in this region. We could not find any measurable sources from potential industry and STP sources based on our tracers.

Second, the authors presented results from PARAFAC analysis in several places (see also Figure 4) and the FDOM composition differences between two samplings. However, there are not additional information related to these analyses, neither component contours nor excitation/emission loadings. The PARAFAC-derived DOM components should be provided along with their detailed and specific Ex/Em values (can be in supplemental if needed).

=> The component contours and excitation/emission loadings will be added in the supplementary section.

Third, Specific goals and scientific questions should be added into the Introduction section. Otherwise, it does not look like a

=> We will more clearly describe specific goals and scientific questions in the revised version.

Fourth, are there any additional parameters/evidence to support the industrial pollution or excess DOC during sampling? Similar to $\delta$13C values, specific C/N ratios should be provided in the abstract so that the reader can easily compare data between different sampling years.

=> As mentioned above, we could not find any sources from industries and STP based on $\delta$13C-DOC, FDOM, and C/N ratio tracers, although we suspected such source inputs before this study was conducted. We will add specific C/N ratio values in the abstract.

Lastly, similar to DOC concentrations, $\delta$13C values in 2016 are somewhat highly variable within small salinity range. What are the possible reasons causing this abrupt change? FDOM, on the other hand, seemed to vary consistently with salinity. Then, the question is that are FDOM nor related to the bulk DOC in the study area or from polluted DOC sources?

=> Yes, this is an important finding of our study. We found increased DOC concentrations at stations showing decreased $\delta$13C-DOC values. This is attributed to artificial island-seawater interactions in addition to terrestrial source inputs from small streams. We will add more information on the artificial island constructed during 2015-16 and the interpretation of this terrestrial source. The good correlations between salinities and FDOM (humic), relative to DOC, are commonly observed since freshwater is the

dominant source for FDOM (humic). However, there is significant marine DOC source (without adding humic FDOM) in the ocean. So, this decoupling is a common phenomenon in coastal waters.

---

## Referee Comment (RC2) · Anonymous Referee #2 · 16 Aug 2019

Review of bg-2019-229 Authors use DOM fluorescence, $\delta$13C ratios, and C:N ratios in an attempt to apportion sources of DOM to Masan Bay, near a heavily industrialized region of Korea. Two samplings were conducted in 2011 and again 2016, axial transects along the bay. Results are presented largely as geospatial plots of parameters and against salinity in a binary conservative mixing focused analysis. Over these 2 samplings, 3 groups of DOM were identified by visual inspection of plots against salinity. Discussion delves in to mixing and potential inclusion of non-conservative sources and concludes that an urban influence was not definable and that local primary production

likely explained the general non-conservative behavior of $\delta$13C and C:N whereas humic fluorescence was largely conservative. The topic of the manuscript is relevant for BGD but the data presentation and analysis need work. The data are certainly interesting and comprised of measurements that are now being combined to understand better the sources and cycling of DOM in coastal waters beyond any one or two of these measurements alone. So the data are solid and appear to provide some insight into this particular region. I have the following suggestions the authors might consider to improve the manuscript. Title – Barring any new insight from further data analysis, it is misleading to have "heavy industrialized cities" in the title. There is no evidence provided and discussed that an urbanization effect was found, only speculation. I think further work is needed on this argument for urban inputs. Writing – overall the writing is good but there are many awkward or unclear phrasings which should be revised to improve readability and clarity. Figure 2 and 3 would benefit from a border around each panel (map) in each figure. **Overall I thought the figures were very good. Data analysis would be improved by biplots beyond property vs salinity. For example, the classic $\delta$13C vs C:N plot could be done (see Lamb, A. L., Wilson, G. P., & Leng, M. J. (2006). A review of coastal palaeoclimate and relative sea-level reconstructions using $\delta$13C and C/N ratios in organic material. Earth-Science Reviews, 75(1-4), 29-57.) to clarify a key uncertainty in the manuscript which is determining inputs of urban runoff DOM vs in-situ generation of phytoplankton DOM. One could imagine $\delta$13C vs FDOMp biplot may elucidate the urban source. Specific comments (line number indicated): L60 – Phytoplankton $\delta$13C values are based on the value of the C they fix; the range specified is for marine phytoplankton. This point should be clarified and considered in lieu of the production in the estuary. L75 – Sentence should be the concluding sentence of the preceding paragraph or otherwise this point should be expanded upon. For example, has there been no prior work on DOM in the region? What are the probable sources of urban DOM that could confound a simple binary mixing analysis? L84 – From satellite imagery, it appears this bay is in a mountainous region, but are there any salt marsh inputs to the bay? This is important because of the effect that C4 plants

such as Spartina might have on DOM inputs. L103 – I am confused by DIN as "inorganic nutrients". First, no nutrient data are shown. Second, DIN typically refers to dissolved inorganic nitrogen (sum of NH4+, NO2-/NO3-). Please clarify. L112 – "qualitatively" instead of "entirely" L136 – equivalents NOTE: More information is needed about the PARAFAC analysis; split-half validated spectra; Plots of the components and distribution of components across stations. Please test the components against the OpenFluor database for matches with other coastal waters. Otherwise does it matter to do PARAFAC? What do BIX and HIX and other derived parameters from fluorescence provide that might obviate the need for PARAFAC? L149: give values when specifying these maxima and minima L156: EEM "PARAFAC" analyses L174: By how much does the freshwater end member vary in its DOC concentration? A freshwater input at 300 $\mu$M DOC could produce conservative mixing patterns (just freshwater and seawater mixing) with a changing freshwater end member value. L177: Explain what is meant here in more detail L180: Seems to argue for multicomponent mixing models L182: Please clarify the evidence for this L188: period missing or otherwise this needs revising to clarify L196: what does "relatively well" mean? Please be specific. Only 2 points fall on the mixing line. L197: -34‰ – is this meant to be terrestrial? It is too depleted a value without some reference to the riverine input or other terrestrial runoff. However, it is possible to be riverine or estuarine phytoplankton with DIC values <-5‰ L215: This is not convincing and the implications of urbanization need to be thought through some more. I would perhaps argue that phytoplankton DOM, enabled by nutrient runoff from land, is the major effect on DOM rather than specifying some non-quantifiable (i.e., according to the manuscript, the data do not exist) urbanized DOM input. L218: Are these tidal creeks with marsh/wetland habitat? How much DOM do they export? L233: What does "natural level" mean? L245-247: Why?? Support this final point; I don't understand how the authors arrive at this conclusion. L251: high and lower than what? L253: This is not correct; refractory nature of DOM cannot be determined by C:N ratios L266: This statement is very clear and summarizes what should be made clearer in the discussion. Tie together these points in the Discussion and the manuscript will be

far more convincing based on the results. I think this is where the biplots I mention above could be very useful. L267: Unclear; please explain in the discussion how the island can influence DOM. L272: I don't understand this last statement in context of this study. Please revise.

————————————————

---

## Author Comment (AC2) · 4 Sep 2019

Review of bg-2019-229 Authors use DOM fluorescence, δ13C ratios, and C:N ratios in an attempt to apportion sources of DOM to Masan Bay, near a heavily industrialized region of Korea. Two samplings were conducted in 2011 and again 2016, axial transects along the bay. Results are presented largely as geospatial plots of parameters and against salinity in a binary conservative mixing focused analysis. Over these 2 samplings, 3 groups of DOM were identified by visual inspection of plots against salinity. Discussion delves in to mixing and potential inclusion of non-conservative sources and concludes that an urban influence was not definable and that local primary likely explained the general non-conservative behavior of $\delta^{13}C$ and C:N whereas humic fluorescence was largely conservative. The topic of the manuscript is relevant for BGD but the data presentation and analysis need work. The data are certainly interesting and comprised of measurements that are now being combined to understand better the sources and cycling of DOM in coastal waters beyond any one or two of these measurements alone. So the data are solid and appear to provide some insight into this particular region. I have the following suggestions the authors might consider to improve the manuscript.

=> Thank you for the constructive and valuable comments.

Title – Barring any new insight from further data analysis, it is misleading to have "heavy industrialized cities" in the title. There is no evidence provided and discussed that an urbanization effect was found, only speculation. I think further work is needed on this argument for urban inputs

=> Yes, we will omit the term "heavy industrialized cities" since the title can mislead our major finding. Although we originally expected to have STP sources, these urban/anthropogenic sources were not measurable. We changed the title to "Tracing terrestrial versus marine sources of dissolved organic carbon occurring in a coastal bay using stable carbon isotopes".

Writing – overall the writing is good but there are many awkward or unclear phrasings which should be revised to improve readability and clarity.

=> We will improve readability and clarity through a native editor.

Figure 2 and 3 would benefit from a border around each panel (map) in each figure. **Overall I thought the figures were very good.

=> will change as suggested.

Data analysis would be improved by biplots beyond property vs salinity. For example, the classic $\delta^{13}C$ vs C:N plot could be done (see Lamb, A. L., Wilson, G. P., & Leng, M. J. (2006). A review of coastal palaeoclimate and relative sea-level reconstructions using δ13C and C/N

ratios in organic material. Earth-Science Reviews, 75(1-4), 29-57.) to clarify a key uncertainty in the manuscript which is determining inputs of urban runoff DOM vs in-situ generation of phytoplankton DOM. One could imagine δ13C vs FDOMp biplot may elucidate the urban source.

=> Thank you for your valuable comments. The plot between $\delta^{13}C$-DOC values and C/N ratios will be added as shown below. The plot further supports that the source of Group1 is mainly influenced by freshwater DOC and the source of Group2 is from marine phytoplankton. The source of Group3 seems to be associated with C3 terrestrial plants, although the specific source is unknown. As shown below, we could not determine the urban source from the $\delta^{13}C$ vs FDOM$_P$ biplot, which was not included in the revised version.

[Figure]

Specific comments (line number indicated):

L60 – Phytoplankton $\delta^{13}C$ values are based on the value of the C they fix; the range specified is for marine phytoplankton. This point should be clarified and considered in lieu of the production in the estuary.

=> will clarify this as "marine phytoplankton" instead of phytoplankton. "those derived from marine phytoplankton range from −18 to −22‰ (Kelley et al., 1998; Coffin and Cifuentes, 1999)."

L75 – Sentence should be the concluding sentence of the preceding paragraph or otherwise this point should be expanded upon. For example, has there been no prior work on DOM in the region? What are the probable sources of urban DOM that could confound a simple binary mixing analysis?

=> We will describe specific goals and scientific questions more clearly in the revised version: "Masan bay is surrounded by cities with thousands of industrial plants and a population of 1.1 million. In association with large anthropogenic nutrient loading, this area has been recognized as a highly eutrophic embayment (Lee and Min, 1990; Yoo, 1991; Hong et al., 2010). The development of red tides and hypoxic water mass in the bottom layer has occurred annually in the spring and summer seasons (Cho et al., 1998; Lee et al., 2009). In addition, there is a sewage treatment plant (STP) as a point source that manages domestic and industrial wastewater of

Masan and Changwon cities. Therefore, in this study, we attempted to use $\delta^{13}$C-DOC, FDOM, and DOC/DON ratios to differentiate different sources and characteristics of DOM in Masan bay waters which have complicated DOM sources."

L84 – From satellite imagery, it appears this bay is in a mountainous region, but are there any salt marsh inputs to the bay? This is important because of the effect that C4 plants such as Spartina might have on DOM inputs.

=> The shore is mostly composed of rocks and concrete walls (no salt marshes or large beaches).

L103 – I am confused by DIN as "inorganic nutrients". First, no nutrient data are shown. Second, DIN typically refers to dissolved inorganic nitrogen (sum of NH4+, NO2-/NO3-). Please clarify.

=> will be corrected to "nitrogen" instead of nutrient. We did not display DIN data since DIN (sum of $NH_4^+$, $NO^{2-}/NO^{3-}$) data were just used for calculating dissolve organic nitrogen (DON = TDN (total dissolved nitrogen) - DIN).

L112 – "qualitatively" instead of "entirely"

=> will be corrected as suggested.

L136 – equivalents NOTE: More information is needed about the PARAFAC analysis; split-half validated spectra; Plots of the components and distribution of components across stations. Please test the components against the OpenFluor database for matches with other coastal waters. Otherwise does it matter to do PARAFAC? What do BIX and HIX and other derived parameters from fluorescence provide that might obviate the need for PARAFAC?

=> More information will be added for the component contours and excitation/emission loadings in the revised version (supplementary section). We will compare the components through OpenFluor. Although we attempted to compare HIX and BIX with salinities, they do not show a good trend as Figure 4 (PARAFAC data). So, we use PARAFAC data in this study.

L149: give values when specifying these maxima and minima

=> will add as suggested.

L156: EEM "PARAFAC" analyses

=> will correct as suggested.

L174: By how much does the freshwater end member vary in its DOC concentration? A freshwater input at 300 µM DOC could produce conservative mixing patterns (just freshwater and seawater mixing) with a changing freshwater end member value.

=> DOC in freshwater endmembers (various creeks) in 2011 were in the range of 120-800 µM. In the case of Group 1, the extrapolated endmember concentration of 200 µM in freshwater seems to be conservatively mixed with open ocean water in the bay.

L177: Explain what is meant here in more detail

=> Although we can extrapolate the concentration and isotope value of the freshwater that influences Group 1, we do not know whether this is natural or anthropogenic sources. So, we

state that Group 1 is influenced by natural and/or anthropogenic land sources in the revised version. The detail of this is included in the next section.

L180: Seems to argue for multicomponent mixing models

=> Yes, we point out the excess source over the two-endmember mixing trend. This excess source is discussed in the next section.

L182: Please clarify the evidence for this

=> In this sentence, we just list up any possible sources for high salinity waters. The evidence is clarified in the next section.

L188: period missing or otherwise this needs revising to clarify

=> will be changed as suggested: "Group 1, Group 2 in 2011, and Group 3 in 2016 (Fig. 4a)"

L196: what does "relatively well" mean? Please be specific. Only 2 points fall on the mixing line.

=> Although $\delta^{13}C$ values of Group 1 are slightly heavier than the mixing line, the trend falls into the mixing line within 1.5 ‰. This will be mentioned in the revised version.

L197: -34‰ – is this meant to be terrestrial? It is too depleted a value without some reference to the riverine input or other terrestrial runoff. However, it is possible to be riverine or estuarine phytoplankton with DIC values <-5‰

=> will be corrected to -32‰ for C3 land plant.

L215: This ˙ is not convincing and the implications of urbanization need to be thought through some more. I would perhaps argue that phytoplankton DOM, enabled by nutrient runoff from land, is the major effect on DOM rather than specifying some non-quantifiable (i.e., according to the manuscript, the data do not exist) urbanized DOM input.

=> We conclude that the source of Group 2 is from biological production based on both isotope values and $\delta^{13}C$ versus DOC/DON ratio plots that you suggested. We rephrase this sentence to be read as "this source could originate from in-situ production".

L218: Are these tidal creeks with marsh/wetland habitat? How much DOM do they export?

=> There is no marsh/wetland habitat in Masan Bay. This will be mentioned in the revised version.

L233: What does "natural level" mean?

=> We meant "no excess DOC is observed" at the station near STP according to the mixing line. This will be clarified in the revised version: "This STP appeared to reduce TOC concentrations which are not discernible from the two endmember mixing, as shown in several other estuaries (Abril et al., 2002)."

L245-247: Why?? Support this final point; I don't understand how the authors arrive at this conclusion.

=> We will clarify this in the revised version: "These results suggest that DOC in Group 3 is

influenced by terrestrial DOC sources which include lower FDOM. The plot between $\delta^{13}$C-DOC versus C/N ratio further indicates that the main source of Group 3 is C3 terrestrial plants. Because salinities of Group 3 are high (26-32), high DOC, depleted $\delta^{13}$C-DOC, and high C/N ratio indicate that this water is directly influenced by terrestrial organic matter.

L251: high and lower than what?

=> will be specified as "than the Group 3".

L253: This is not correct; refractory nature of DOM cannot be determined by C:N ratios

=> Yes, C:N ratios and refractory nature are not correlated directly, although DOM which has higher C:N ratios is generally more refractory in the ocean (Andrews et al., 1998). We removed the term "refractory" in the sentence.

L266: This statement is very clear and summarizes what should be made clearer in the discussion. Tie together these points in the Discussion and the manuscript will be far more convincing based on the results. I think this is where the biplots I mention above could be very useful.

=> Thank you.

L267: Unclear; please explain in the discussion how the island can influence DOM.

=> Yes, we will explain about this in the revised version.

L272: I don't understand this last statement in context of this study. Please revise.

=> will clarify in the revised version: "Our results show that the combination of multiple DOM tracers, including $\delta^{13}$C-DOC, FDOM, and DOC/DON ratios, is powerful for discriminating the complicated sources of DOM in coastal waters."

---

## Author Response (AR1)

Lee and Guebuem Kim
Anonymous Referee #1

**Major comments**
This manuscript by Lee et al set out to investigate the sources of dissolved organic matter in a coastal bay surrounded by heavily industrialized cities, Masan Bay, Korea, during two different sampling trips in 2011 and 2016. The authors measured DOC, DON, chromophoric-DOM and stable carbon isotope composition in related samples. They found that the excess DOC was observed in higher-salinity waters (16-21) during 2011, with higher concentrations of protein-like FDOM and lower DOC/DON ratios. In 2016, however, the high DOC waters in high-salinity waters were characterized by low FDOM, more depleted $\delta13C$ values and high C/N ratios.

In general, this is a fairly well written manuscript presenting data on the source changes in DOC and CDOM in a coastal environment during two different sampling years. Overall, it is concise manuscript. Their approach is straightforward and conclusions are based on what the authors observed. Contents presented here are suitable for the journal of bg. I support eventual publication of this manuscript.

Having said that this manuscript needs revisions before acceptance.

=> Thank you for the positive and valuable comments.

First of all, their two samplings in the same study area are over 5 years apart (2011-2016) without other sampling point. In addition, DOC abundances do not vary consistently or significantly with salinity (Fig 4a) although FDOM-H does (Figure 4c). It is very difficult to judge these DOC data since they are from a coastal bay influenced heavily by industrialized. Additional explanation will be helpful.

=> Yes, we included more explanation on oceanographic changes over different sampling years in the revised version (line 105): "The averages of surface water temperature were $30.4 \pm 2.3°C$ in 2011 and $26.5 \pm 0.7°C$ in 2016." We use these multiple tracers in order to determine the main sources amongst such inconsistent and dynamic sources in this region. We could not find any measurable sources from potential industry and STP sources based on our tracers.

Second, the authors presented results from PARAFAC analysis in several places (see also Figure 4) and the FDOM composition differences between two samplings. However, there are not additional information related to these analyses, neither component contours nor excitation/emission loadings. The PARAFAC-derived DOM components should be provided along with their detailed and specific Ex/Em values (can be in supplemental if needed).

=> The component contours and excitation/emission loadings were added in the supplementary.

Third, Specific goals and scientific questions should be added into the Introduction section. Otherwise, it does not look like a

=> We more clearly described specific goals and scientific questions in the revised version (line 76-85). "Masan bay is surrounded by cities with thousands of industrial plants and a population of 1.1 million. In association with large anthropogenic nutrient loading, this area has been recognized as a highly eutrophic embayment (Lee and Min, 1990; Yoo, 1991; Hong et al., 2010). The development of red tides and hypoxic water mass in the bottom layer of the bay has occurred annually in spring and summer (Lee et al., 2009). In addition, there are point sources from sewage treatment plants (STPs), which manage domestic and industrial wastewater from Masan and Changwon cities. Therefore, in this study, we attempted to use $\delta^{13}$C-DOC, FDOM,

and DOC/DON ratios to differentiate various sources and characteristics of DOM in Masan bay waters, which may include many different types of terrestrial/anthropogenic and marine sources of DOM."

Fourth, are there any additional parameters/evidence to support the industrial pollution or excess DOC during sampling? Similar to $\delta^{13}C$ values, specific C/N ratios should be provided in the abstract so that the reader can easily compare data between different sampling years.

=> As mentioned above, we could not find any sources from industries and STP based on $\delta^{13}C$-DOC, FDOM, and C/N ratio tracers, although we suspected such source inputs before this study was conducted. We added specific C/N ratio values in the abstract (line 32 and 34).

Lastly, similar to DOC concentrations, $\delta^{13}C$ values in 2016 are somewhat highly variable within small salinity range. What are the possible reasons causing this abrupt change? FDOM, on the other hand, seemed to vary consistently with salinity. Then, the question is that are FDOM nor related to the bulk DOC in the study area or from polluted DOC sources?

=> Yes, this is an important finding of our study. We found increased DOC concentrations at the stations showing decreased $\delta^{13}C$-DOC values. This is attributed to direct land-seawater interactions based on the plot of $\delta^{13}C$-DOC values versus C/N ratios (Fig. 5). The good correlations between salinities and FDOM (humic), relative to DOC, are commonly observed since freshwater is the dominant source for FDOM (humic). However, there is significant marine DOC source (without adding humic FDOM) in the ocean. So, this decoupling is a common phenomenon in coastal waters.
Review of bg-2019-229 Authors use DOM fluorescence, $\delta 13C$ ratios, and C:N ratios in an attempt to apportion sources of DOM to Masan Bay, near a heavily industrialized region of Korea. Two samplings were conducted in 2011 and again 2016, axial transects along the bay. Results are presented largely as geospatial plots of parameters and against salinity in a binary conservative mixing focused analysis. Over these 2 samplings, 3 groups of DOM were identified by visual inspection of plots against salinity. Discussion delves in to mixing and potential inclusion of non-conservative sources and concludes that an urban influence was not definable and that local primary likely explained the general non-conservative behavior of $\delta^{13}C$ and C:N whereas humic fluorescence was largely conservative. The topic of the manuscript is relevant for BGD but the data presentation and analysis need work. The data are certainly interesting and comprised of measurements that are now being combined to understand better the sources and cycling of DOM in coastal waters beyond any one or two of these measurements alone. So the data are solid and appear to provide some insight into this particular region. I have the following suggestions the authors might consider to improve the manuscript.

=> Thank you for the constructive and valuable comments.

Title – Barring any new insight from further data analysis, it is misleading to have "heavy industrialized cities" in the title. There is no evidence provided and discussed that an

urbanization effect was found, only speculation. I think further work is needed on this argument for urban inputs

=> Yes, we omitted the term "heavy industrialized cities" since the title can mislead our major finding. Although we originally expected to have STP sources, these urban/anthropogenic sources were not measurable. We changed the title to "Tracing terrestrial versus marine sources of dissolved organic carbon in a coastal bay using stable carbon isotopes".

Writing – overall the writing is good but there are many awkward or unclear phrasings which should be revised to improve readability and clarity.

=> We improved readability and clarity through a native editor.

Figure 2 and 3 would benefit from a border around each panel (map) in each figure. **Overall I thought the figures were very good.

=> changed as suggested.

[Figure]

Data analysis would be improved by biplots beyond property vs salinity. For example, the classic $\delta^{13}C$ vs C:N plot could be done (see Lamb, A. L., Wilson, G. P., & Leng, M. J. (2006). A review of coastal palaeoclimate and relative sea-level reconstructions using δ13C and C/N ratios in organic material. Earth-Science Reviews, 75(1-4), 29-57.) to clarify a key uncertainty in the manuscript which is determining inputs of urban runoff DOM vs in-situ generation of phytoplankton DOM. One could imagine δ13C vs FDOMp biplot may elucidate the urban source.

=> Thank you for your valuable comments. The plot between $\delta^{13}C$-DOC values and C/N ratios was added as shown below. The plot further supports that the source of Group1 is mainly influenced by freshwater DOC and the source of Group2 is from marine phytoplankton. The source of Group3 seems to be associated with C3 terrestrial plants, although the specific source is unknown. As shown below, we could not determine the urban source from the $\delta^{13}C$ vs FDOM$_P$ biplot, except anomalous FDOM$_P$ increases by marine production in 2011. Rather than this plot, we verified no inputs of STP or urban sources based on $\delta^{13}C$-DOC, FDOM$_P$, fulvic-like peak, and salinity plots (lines 241-259) in the revised version.

[Figure]

Specific comments (line number indicated):

L60 – Phytoplankton $\delta^{13}C$ values are based on the value of the C they fix; the range specified is for marine phytoplankton. This point should be clarified and considered in lieu of the production in the estuary.

=> clarified this as "marine phytoplankton" instead of phytoplankton. "those derived from marine phytoplankton range from −18 to −22‰ (Kelley et al., 1998; Coffin and Cifuentes, 1999)."

L75 – Sentence should be the concluding sentence of the preceding paragraph or otherwise this point should be expanded upon. For example, has there been no prior work on DOM in the region? What are the probable sources of urban DOM that could confound a simple binary mixing analysis?

=> We described specific goals and scientific questions more clearly in the revised version (line 76-85): "Masan bay is surrounded by cities with thousands of industrial plants and a population of 1.1 million. In association with large anthropogenic nutrient loading, this area has been recognized as a highly eutrophic embayment (Lee and Min, 1990; Yoo, 1991; Hong et al., 2010). The development of red tides and hypoxic water mass in the bottom layer of the bay has occurred annually in spring and summer (Lee et al., 2009). In addition, there are point sources from sewage treatment plants (STPs), which manage domestic and industrial wastewater from Masan and Changwon cities. Therefore, in this study, we attempted to use $\delta^{13}C$-DOC, FDOM, and DOC/DON ratios to differentiate various sources and characteristics of DOM in Masan bay waters, which may include many different types of terrestrial/anthropogenic and marine sources of DOM."

L84 – From satellite imagery, it appears this bay is in a mountainous region, but are there any salt marsh inputs to the bay? This is important because of the effect that C4 plants such as Spartina might have on DOM inputs.

=> The shore is mostly composed of rocks and concrete walls (no salt marshes or large beaches).

L103 – I am confused by DIN as "inorganic nutrients". First, no nutrient data are shown.

Second, DIN typically refers to dissolved inorganic nitrogen (sum of NH4+, NO2-/NO3-). Please clarify.

=> corrected to "nitrogen" instead of nutrient. We did not display DIN data since DIN (sum of $NH_4^+$, $NO_2^-$/$NO_3^-$) data were just used for calculating dissolve organic nitrogen (DON = TDN (total dissolved nitrogen) - DIN).

L112 – "qualitatively" instead of "entirely"

=> corrected.

L136 – equivalents NOTE: More information is needed about the PARAFAC analysis; split-half validated spectra; Plots of the components and distribution of components across stations. Please test the components against the OpenFluor database for matches with other coastal waters. Otherwise does it matter to do PARAFAC? What do BIX and HIX and other derived parameters from fluorescence provide that might obviate the need for PARAFAC?

=> More information was added for the component contours and excitation/emission loadings in the supplementary. We compared the components through OpenFluor. Component 1 was matched with marine humic substances (Ex = 290–320 nm, Em = 370–420 nm). Component 2 resembles terrestrial humic substances (Ex = 320–360 nm, Em = 420–460 nm). Component 3 was matched with protein-like component (Ex = 275–300 nm, Em = 340–360 nm). Although we attempted to compare HIX and BIX with salinities, they do not show a good trend as Figure 4 (PARAFAC data). So, we use PARAFAC data in this study.

L149: give values when specifying these maxima and minima

=> added as suggested.

L156: EEM "PARAFAC" analyses

=> corrected as suggested.

L174: By how much does the freshwater end member vary in its DOC concentration? A freshwater input at 300 µM DOC could produce conservative mixing patterns (just freshwater and seawater mixing) with a changing freshwater end member value.

=> DOC in freshwater endmembers (various creeks) in 2011 were in the range of 120-800 µM. In the case of Group 1, the extrapolated endmember concentration of 200 µM in freshwater seems to be conservatively mixed with open ocean water in the bay.

L177: Explain what is meant here in more detail

=> Although we can extrapolate the concentration and isotope value of the freshwater that influences Group 1, we do not know whether this is natural or anthropogenic sources. So, we state that Group 1 is influenced by natural and/or anthropogenic land sources in the revised version. The detail of this is included in the next section.

L180: Seems to argue for multicomponent mixing models

=> Yes, we point out the excess source over the two-endmember mixing trend. This excess source is discussed in the next section.

L182: Please clarify the evidence for this

=> In this sentence, we just list up any possible sources for high salinity waters. The evidence is clarified in the next section.

L188: period missing or otherwise this needs revising to clarify

=> changed as suggested: "Group 1, Group 2 in 2011, and Group 3 in 2016 (Fig. 4a)"

L196: what does "relatively well" mean? Please be specific. Only 2 points fall on the mixing line.

=> Although $\delta^{13}$C values of Group 1 are slightly heavier than the mixing line, the trend falls into the mixing line within 1.5 ‰. This is mentioned in the revised version (line 204).

L197: -34‰ – is this meant to be terrestrial? It is too depleted a value without some reference to the riverine input or other terrestrial runoff. However, it is possible to be riverine or estuarine phytoplankton with DIC values <-5‰

=> corrected to -32‰ for C3 land plant.

L215: This ˙ is not convincing and the implications of urbanization need to be thought through some more. I would perhaps argue that phytoplankton DOM, enabled by nutrient runoff from land, is the major effect on DOM rather than specifying some non-quantifiable (i.e., according to the manuscript, the data do not exist) urbanized DOM input.

=> We conclude that the source of Group 2 is from biological production based on both isotope values and $\delta^{13}$C versus DOC/DON ratio plots that you suggested. We rephrased this sentence to be read as "the excess DOC of Group 2 is produced by marine phytoplankton" (line 215-216).

L218: Are these tidal creeks with marsh/wetland habitat? How much DOM do they export?

=> There is no marsh/wetland habitat in Masan Bay. This will be mentioned in the revised version.

L233: What does "natural level" mean?

=> We meant "no excess DOC is observed" at the station near STP according to the mixing line. This is clarified in the revised version (line 254-256): "This STP appears to reduce TOC concentrations to a level that cannot influence the DOC concentrations resulting from the other mixing sources, as shown in several other estuaries (Abril et al., 2002)"

L245-247: Why?? Support this final point; I don't understand how the authors arrive at this conclusion.

=> clarified this in the revised version: "The plot of $\delta^{13}$C-DOC values versus C/N ratios indicates that the excess DOC of Group 3 is from C3 terrestrial plants through direct land-seawater interactions based on the fact that the excess DOC occurred in high-salinity (26–32) waters (Fig. 5)."

L251: high and lower than what?

=> specified as "than the Group 3".

L253: This is not correct; refractory nature of DOM cannot be determined by C:N ratios

=> Yes, C:N ratios and refractory nature are not correlated directly, although DOM which has higher C:N ratios is generally more refractory in the ocean (Andrews et al., 1998). We removed the term "refractory" in the sentence.

L266: This statement is very clear and summarizes what should be made clearer in the discussion. Tie together these points in the Discussion and the manuscript will be far more convincing based on the results. I think this is where the biplots I mention above could be very useful.

=> Thank you.

L267: Unclear; please explain in the discussion how the island can influence DOM.

=> Yes, we explained about this in the revised version. "The excess DOC concentrations in the third group in high salinity waters in 2016 seemed to be produced by direct interaction between land and seawater based on more depleted $\delta^{13}$C-DOC values (−28.8‰ and −21.1‰), low FDOM concentrations, and high C/N ratios."

L272: I don't understand this last statement in context of this study. Please revise.

=> clarified in the revised version: "Our results show that the combination of multiple DOM tracers, including $\delta^{13}$C-DOC, FDOM, and DOC/DON ratios, is powerful for discriminating the complicated sources of DOM occurring in coastal waters."

---

## Author Response (AR2)

We would like to thank the editor and the two referees for their constructive reviews and valuable suggestions on this manuscript. We have made every attempt to fully address these comments in the revised version. As the editor suggested, we carried out revisions in response to every point and refer to how the enhanced explanations have been incorporated into the text. The revision of the manuscript is thoroughly done, and all such details are provided in our new response below.

We hope that the changes made into the revised version are satisfactory, and the manuscript is now acceptable for publication in *Biogeosciences*.

Thank you again for your consideration of this manuscript.

**Referee #1**
**Major comments**
This manuscript by Lee et al set out to investigate the sources of dissolved organic matter in a coastal bay surrounded by heavily industrialized cities, Masan Bay, Korea, during two different sampling trips in 2011 and 2016. The authors measured DOC, DON, chromophoric-DOM and stable carbon isotope composition in related samples. They found that the excess DOC was observed in higher-salinity waters (16-21) during 2011, with higher concentrations of protein-like FDOM and lower DOC/DON ratios. In 2016, however, the high DOC waters in high-salinity waters were characterized by low FDOM, more depleted δ13C values and high C/N ratios.
In general, this is a fairly well written manuscript presenting data on the source changes in DOC and CDOM in a coastal environment during two different sampling years. Overall, it is concise manuscript. Their approach is straightforward and conclusions are based on what the authors observed. Contents presented here are suitable for the journal of bg. I support eventual publication of this manuscript.
Having said that this manuscript needs revisions before acceptance.
=> Thank you for the positive and valuable comments.
First of all, their two samplings in the same study area are over 5 years apart (2011-2016) without other sampling point. In addition, DOC abundances do not vary consistently or significantly with salinity (Fig 4a) although FDOM-H does (Figure 4c). It is very difficult to judge these DOC data since they are from a coastal bay influenced heavily by industrialized. Additional explanation will be helpful.
=> Yes, we included more explanation on oceanographic changes over different sampling years in the revised version (lines 101-103, 108-109). We use these multiple tracers in order to determine the main sources amongst such inconsistent and dynamic sources in this region. We could not find any measurable DOC inputs from industry and STP sources based on our multiple tracers.
Second, the authors presented results from PARAFAC analysis in several places (see also Figure 4) and the FDOM composition differences between two samplings. However, there are not additional information related to these analyses, neither component contours nor excitation/emission loadings. The PARAFAC-derived DOM components should be provided

along with their detailed and specific Ex/Em values (can be in supplemental if needed).

=> The information on "component contours and excitation/emission loadings" was added in the supplementary.

=> In "Methods" and "Results and Discussion" sections, we add more information on the component details and specific Ex/Em values and the methods: "the three components (C1-C3) were validated by split-half analysis (Figs. S1 and S2) (line 151)". "EEMs contour plots and split-half validation results of three components are shown in the supplementary (Fig. S1 and S2). Based on the comparison with the data in the OpenFluor (Murphy et al., 2014), Component 1 (FDOM$_H$, Ex/Em = 322/405 nm) is associated with a terrestrial humic-like component (Liu et al., 2019; Dalmagro et al., 2019; Chen et al., 2016) . Component 2 (FDOM$_M$, Ex/Em = 386/450 nm) is also associated with an allochthonous humic-like component (Wünsch et al., 2017). Component 3 (FDOM$_P$, Ex/Em = 280/330 nm) is associated with a protein-like component, which is a product of microbial processes (Liu et al., 2019; Murphy et al., 2011; Osburn et al., 2011)." (lines 167-175).

Third, Specific goals and scientific questions should be added into the Introduction section. Otherwise, it does not look like a

=> We more clearly described specific goals and scientific questions in the revised version: "Masan bay is surrounded by cities with thousands of industrial plants and a population of 1.1 million. In association with large anthropogenic nutrient loading, this area has been recognized as a highly eutrophic embayment (Lee and Min, 1990; Yoo, 1991; Hong et al., 2010). The development of red tides and hypoxic water mass in the bottom layer of the bay has occurred annually in spring and summer (Lee et al., 2009). In addition, there are potential point sources from sewage treatment plants (STPs), which manage domestic and industrial wastewater from Masan and Changwon cities. Therefore, Masan Bay is a suitable place to trace and characterize various DOM sources in seawater. However, only a few studies have been conducted to determine DOM sources in this bay. Lee et al. (2011) revealed the origins of sewage and organic matter using dissolved sterols in Masan Bay. They reported that the water samples from the creeks, inner bay, and near STP were affected by sewage sources. Oh et al. (2017) showed that the excess DOC in bay water is produced by phytoplankton production. Therefore, in this study, we attempted to use multiple tracers including $\delta^{13}$C-DOC, FDOM, and DOC/DON ratios to determine major sources and characteristics of DOM in Masan bay waters." (lines 77-90)

Fourth, are there any additional parameters/evidence to support the industrial pollution or excess DOC during sampling? Similar to $\delta^{13}$C values, specific C/N ratios should be provided in the abstract so that the reader can easily compare data between different sampling years.

=> As mentioned above, we could not find any measurable source inputs from industries and STP based on $\delta^{13}$C-DOC, FDOM, and C/N ratio tracers, although we suspected such source inputs originally. We added information on "specific C/N ratio values" in the abstract (lines 31 and 33).

Lastly, similar to DOC concentrations, $\delta^{13}$C values in 2016 are somewhat highly variable within small salinity range. What are the possible reasons causing this abrupt change? FDOM, on the other hand, seemed to vary consistently with salinity. Then, the question is that are FDOM nor related to the bulk DOC in the study area or from polluted DOC sources?

=> Yes, this is an important finding of our study. We found increased DOC concentrations at the stations showing decreased $\delta^{13}$C-DOC values. This is attributed to direct land-seawater interactions based on the plot of $\delta^{13}$C-DOC values versus C/N ratios (Fig. 5). We mention about this in the revised version: "The plot of $\delta^{13}$C-DOC values versus C/N ratios indicates that the excess DOC of Group 3 is from C3 terrestrial plants through direct land (including the possible

sources from a newly-built artificial island)-seawater interactions, based on the fact that the excess DOC occurred in high-salinity (26–32) waters (Fig. 5a)." (lines 232-235)
=> The significant correlation between salinities and $FDOM_H$ is commonly observed in coastal waters since freshwater is the dominant source for $FDOM_H$. However, there are various marine DOC sources that do not add humic FDOM in the ocean. We describe that "We observed the decoupling between DOC and $FDOM_H$ because $FDOM_H$ is not the major portion of DOC in this bay, except M4-1 station."(lines 247-248)

**Referee #2**

Review of bg-2019-229 Authors use DOM fluorescence, δ13C ratios, and C:N ratios in an attempt to apportion sources of DOM to Masan Bay, near a heavily industrialized region of Korea. Two samplings were conducted in 2011 and again 2016, axial transects along the bay. Results are presented largely as geospatial plots of parameters and against salinity in a binary conservative mixing focused analysis. Over these 2 samplings, 3 groups of DOM were identified by visual inspection of plots against salinity. Discussion delves in to mixing and potential inclusion of non-conservative sources and concludes that an urban influence was not definable and that local primary likely explained the general non-conservative behavior of $\delta^{13}C$ and C:N whereas humic fluorescence was largely conservative. The topic of the manuscript is relevant for BGD but the data presentation and analysis need work. The data are certainly interesting and comprised of measurements that are now being combined to understand better the sources and cycling of DOM in coastal waters beyond any one or two of these measurements alone. So the data are solid and appear to provide some insight into this particular region. I have the following suggestions the authors might consider to improve the manuscript.
=> Thank you for the constructive and valuable comments.
Title – Barring any new insight from further data analysis, it is misleading to have "heavy industrialized cities" in the title. There is no evidence provided and discussed that an urbanization effect was found, only speculation. I think further work is needed on this argument for urban inputs
=> Yes, we omitted the term "heavy industrialized cities" since the title can mislead our major finding. Although we originally expected to have STP sources, these urban/anthropogenic sources were not measurable. We changed the title to "Tracing terrestrial versus marine sources of dissolved organic carbon in a coastal bay using stable carbon isotopes".
Writing – overall the writing is good but there are many awkward or unclear phrasings which should be revised to improve readability and clarity.
=> We improved readability and clarity through a native editor.
Figure 2 and 3 would benefit from a border around each panel (map) in each figure. **Overall I thought the figures were very good.
=> changed as suggested.

[Figure]

Data analysis would be improved by biplots beyond property vs salinity. For example, the classic $\delta^{13}C$ vs C:N plot could be done (see Lamb, A. L., Wilson, G. P., & Leng, M. J. (2006). A review of coastal palaeoclimate and relative sea-level reconstructions using δ13C and C/N ratios in organic material. Earth-Science Reviews, 75(1-4), 29-57.) to clarify a key uncertainty in the manuscript which is determining inputs of urban runoff DOM vs in-situ generation of phytoplankton DOM. One could imagine δ13C vs FDOMp biplot may elucidate the urban source.

=> Thank you for your valuable comments. The suggested plots between $\delta^{13}C$-DOC values and C/N ratios are shown below and added in the revised version (Figure 5a). We included more explanation on this in the revised version (lines 214-215, 222-224, and 232-235). The plot further supports that the source of Group1 is mainly influenced by freshwater DOC and the source of Group2 is from marine phytoplankton. The source of Group3 seems to be associated with C3 terrestrial plants, although the specific source is unknown.

=> In general, anomalously high FDOM$_P$ was observed for anthropogenic sources (Coble, 1996; Baker et al., 2003). The $\delta^{13}C$ values of sewage effluents generally ranged from –22‰ to –28.5‰ (Andrews et al., 1998; Barros et al., 2010), and those of STP effluents ranged from –24‰ to –28‰ (Griffith et al., 2009). The $\delta^{13}C$ vs FDOM$_P$ plot (Fig. 5b) shows that there was no increase in FDOM$_P$ concentrations for samples which had depleted $\delta^{13}C$ values. Thus, we conclude that there was no significant DOC input from untreated sewage or STP sources in this bay. This is mentioned in the revised version (lines 266-272).

[Figure]

Specific comments (line number indicated):
L60 – Phytoplankton $\delta^{13}C$ values are based on the value of the C they fix; the range specified is for marine phytoplankton. This point should be clarified and considered in lieu of the production in the estuary.

=> clarified this as "marine phytoplankton" instead of phytoplankton. "those derived from marine phytoplankton range from −18 to −22‰ (Kelley et al., 1998; Coffin and Cifuentes, 1999)."

L75 – Sentence should be the concluding sentence of the preceding paragraph or otherwise this point should be expanded upon. For example, has there been no prior work on DOM in the region? What are the probable sources of urban DOM that could confound a simple binary mixing analysis?

=> We added more statements on (1) description of Masan Bay, (2) study objectives, (3) prior work on DOM in this bay, and (4) probable sources of urban DOM that could confound a simple binary mixing analysis in the revised version: "Masan bay is surrounded by cities with thousands of industrial plants and a population of 1.1 million. In association with large anthropogenic nutrient loading, this area has been recognized as a highly eutrophic embayment (Lee and Min, 1990; Yoo, 1991; Hong et al., 2010). The development of red tides and hypoxic water mass in the bottom layer of the bay has occurred annually in spring and summer (Lee et al., 2009). In addition, there are potential point sources from sewage treatment plants (STPs), which manage domestic and industrial wastewater from Masan and Changwon cities. Therefore, Masan Bay is a suitable place to trace and characterize various DOM sources in seawater. However, only a few studies have been conducted to determine DOM sources in this bay. Lee et al. (2011) revealed the origins of sewage and organic matter using dissolved sterols in Masan Bay. They reported that the water samples from the creeks, inner bay, and near STP were affected by sewage sources. Oh et al. (2017) showed that the excess DOC in bay water is produced by phytoplankton production. Therefore, in this study, we attempted to use multiple tracers including $\delta^{13}$C-DOC, FDOM, and DOC/DON ratios to determine major sources and characteristics of DOM in Masan bay waters." (lines 77-90)

L84 – From satellite imagery, it appears this bay is in a mountainous region, but are there any salt marsh inputs to the bay? This is important because of the effect that C4 plants such as Spartina might have on DOM inputs.

=> The shore is mostly composed of rocks and concrete walls (no salt marshes or large beaches). This is mentioned in the revised version (lines 200-201).

L103 – I am confused by DIN as "inorganic nutrients". First, no nutrient data are shown. Second, DIN typically refers to dissolved inorganic nitrogen (sum of NH4+, NO2-/NO3-). Please clarify.

=> corrected to "nitrogen" instead of nutrient (line 113). We did not display DIN data since DIN (sum of $NH_4^+$, $NO^{2-}/NO^{3-}$) data were just used for calculating dissolve organic nitrogen (DON = TDN (total dissolved nitrogen) - DIN).

L112 – "qualitatively" instead of "entirely"

=> corrected.

L136 – equivalents NOTE: More information is needed about the PARAFAC analysis; split-half validated spectra; Plots of the components and distribution of components across stations. Please test the components against the OpenFluor database for matches with other coastal waters. Otherwise does it matter to do PARAFAC? What do BIX and HIX and other derived parameters from fluorescence provide that might obviate the need for PARAFAC?

=> More information was added for the component contours and excitation/emission loadings in the supplementary.

=> We added the statements: "Based on the comparison with the data in the OpenFluor (Murphy et al., 2014), Component 1 ($FDOM_H$, Ex/Em = 322/405 nm) is associated with a terrestrial humic-like component (Liu et al., 2019; Dalmagro et al., 2019; Chen et al., 2016) .

Component 2 (FDOM$_M$, Ex/Em = 386/450 nm) is also associated with an allochthonous humic-like component (Wünsch et al., 2017). Component 3 (FDOM$_P$, Ex/Em = 280/330 nm) is associated with a protein-like component, which is a product of microbial processes (Liu et al., 2019; Murphy et al., 2011; Osburn et al., 2011)." (lines 167-175)

=> Although we attempted to compare HIX and BIX with salinities, they did not show a good trend. Thus, we use Figure 4 (PARAFAC data) in this study.

L149: give values when specifying these maxima and minima

=> added as suggested.

L156: EEM "PARAFAC" analyses

=> corrected as suggested.

L174: By how much does the freshwater end member vary in its DOC concentration? A freshwater input at 300 µM DOC could produce conservative mixing patterns (just freshwater and seawater mixing) with a changing freshwater end member value.

=> DOC in freshwater endmembers (various creeks) in 2011 were in the range of 120-800 µM. In the case of Group 1, the extrapolated endmember concentration of 200 µM in freshwater seems to be conservatively mixed with the open ocean water in the bay.

L177: Explain what is meant here in more detail

=> Although we can extrapolate the concentration and isotope value of the freshwater that influences Group 1, we do not know whether it is natural or anthropogenic. We omitted this sentence and changed the sentence to "The potential sources of excess DOC occurring in this bay water may include terrestrial freshwater in creeks, STP water, direct land-seawater interaction, and in-situ biological production. The creek water may also include various anthropogenic sources (i.e., industrial, agricultural, and domestic sewage) as well as natural land sources. There are no salt-marsh or wetland habitat in Masan Bay" (lines 197-201).

L180: Seems to argue for multicomponent mixing models

=> Yes, we point out the excess source over the two-endmember mixing trend. This excess source is discussed in the next section (lines 218-224).

L182: Please clarify the evidence for this

=> In this sentence, we just list up any possible sources for high salinity waters. The evidence is clarified in the next section (lines 218-224). We omitted this sentence and changed the sentence to "The potential sources of excess DOC occurring in this bay water may include terrestrial freshwater in creeks, STP water, direct land-seawater interaction, and in-situ biological production. The creek water may also include various anthropogenic sources (i.e., industrial, agricultural, and domestic sewage) as well as natural land sources. There are no salt-marsh or wetland habitat in Masan Bay." (lines 197-201)

L188: period missing or otherwise this needs revising to clarify

=> changed as suggested: "Group 1, Group 2 in 2011, and Group 3 in 2016 (Fig. 4a)"

L196: what does "relatively well" mean? Please be specific. Only 2 points fall on the mixing line.

=> We changed the statement to "The $\delta^{13}$C values of Group 1 fall into the mixing line or were slightly heavier than the mixing line within 1.5 ‰, indicating the conservative mixing between the terrestrial C3 land plant (−23‰ to −32‰) in freshwater and the open ocean seawater. The slightly heavier values could be produced by in-situ biological mixing during the mixing processes. (lines 210-215)"

L197: -34‰ – is this meant to be terrestrial? It is too depleted a value without some reference to the riverine input or other terrestrial runoff. However, it is possible to be riverine or estuarine phytoplankton with DIC values <-5‰

=> corrected to -32‰ for C3 land plant (Deines, 1980).

L215: This ˙ is not convincing and the implications of urbanization need to be thought through some more. I would perhaps argue that phytoplankton DOM, enabled by nutrient runoff from land, is the major effect on DOM rather than specifying some non-quantifiable (i.e., according to the manuscript, the data do not exist) urbanized DOM input.

=> We conclude that the source of Group 2 is from biological production based on both isotope values and $\delta^{13}$C versus DOC/DON ratio plots that you suggested. We rephrased this sentence to be read as "the excess DOC of Group 2 is produced by marine phytoplankton" (lines 222-224).

L218: Are these tidal creeks with marsh/wetland habitat? How much DOM do they export?

=> There is no marsh/wetland habitat in Masan Bay. This will be mentioned in the revised version (lines 200-201).

L233: What does "natural level" mean?

=> We meant "no excess DOC is observed" at the station near STP according to the mixing line. This is clarified in the revised version: "This STP appears to reduce TOC concentrations to a level that cannot influence the DOC concentrations resulting from the other mixing sources, as shown in several other estuaries (Abril et al., 2002)." (lines 263-265)

L245-247: Why?? Support this final point; I don't understand how the authors arrive at this conclusion.

=> We clarified this in the revised version using a $\delta^{13}$C-DOC vs. C/N ratio plot: "The plot of $\delta^{13}$C-DOC values versus C/N ratios indicates that the excess DOC of Group 3 is from C3 terrestrial plants through direct land (including the possible sources from a newly-built artificial island)-seawater interactions, based on the fact that the excess DOC occurred in high-salinity (26–32) waters (Fig. 5a)." (lines 232-235)

L251: high and lower than what?

=> The sentence including this phrase was omitted in the revised version.

L253: This is not correct; refractory nature of DOM cannot be determined by C:N ratios

=> Yes, C:N ratios and refractory nature are not correlated directly, although DOM which has higher C:N ratios is generally more refractory in the ocean (Andrews et al., 1998). We removed the term "refractory" in the sentence.

L266: This statement is very clear and summarizes what should be made clearer in the discussion. Tie together these points in the Discussion and the manuscript will be far more convincing based on the results. I think this is where the biplots I mention above could be very useful.

=> Thank you.

L267: Unclear; please explain in the discussion how the island can influence DOM.

=> Since the contribution of artificial island constructions is unclear at this stage, we omitted the term "the island" in the revised version. We clarified this in the revised version: "The excess DOC concentrations in the third group in high salinity waters in 2016 seemed to be produced by direct interaction between land and seawater based on more depleted $\delta^{13}$C-DOC values (−28.8‰ and −21.1‰), low FDOM concentrations, and high C/N ratios." (lines 283-285)

L272: I don't understand this last statement in context of this study. Please revise.

=> clarified in the revised version: "Our results show that the combination of multiple DOM tracers, including $\delta^{13}$C-DOC, FDOM, and DOC/DON ratios, is powerful for discriminating the complicated sources of DOM occurring in coastal waters." (lines 285-288)

---

## Author Response (AR3)

We would like to thank for your valuable suggestions on this manuscript again. We revised the manuscript as shown below.

1) I still find the research objectives in the introduction rather vague. Please lay these out as research questions or hypotheses that you have tested, and which map directly onto the main findings that you summarise in your conclusions. Further, please make it clear how your study moves the field forward (perhaps focus on the combination of indices that you have used), especially in comparison to the previous studies of organic pollutants etc., and in terms of all such estuaries rather than just your study site.

=> improved the introduction section as suggested.

2) Where you report statistical test results - r2 values - please name the text you used (linear regression?).

=> changed as suggested (lines 512-513).

3) Where you state the isotopic values DOC end members please provide references.

=> added as suggested (line 215, line 216, line 219, and line 263).

[revised manuscript text omitted]